# Nano-Grain Ni/ZrO₂ Functional Gradient Coating Fabricated by Double Pulses Electrodeposition with Enhanced High Temperature Corrosion Performance

**Wen Ge \*, Taisong He, Meijiao Wang and Jiamei Li**

Engineering Research Center of Nano-Geomaterials of Ministry of Education, Faculty of Materials Science and Chemistry, China University of Geosciences, Wuhan 430074, China; Hetaisong1994@163.com (T.H.); Mecho@cug.edu.cn (M.W.); Lijiamei1902@163.com (J.L.)

**\*** Correspondence: gewen@cug.edu.cn

**Abstract:** Functional gradient materials (FGM) have many excellent properties, and high-performance gradient coating exhibits good prospective application. In this paper, the nano-grain Ni/ZrO₂ gradient coating was prepared by double pulse electrodeposition (BP). The surface morphology, crystal structure and electrochemical corrosion resistance of the nano-grain Ni/ZrO₂ coating and Ni coating, annealed at different temperatures (400–800 °C), have been compared. In the vertical direction to the substrate surface, the content of ZrO₂ increases from 0% to 34.99%. X-ray diffraction (XRD) revealed that the average crystal size of Ni/ZrO₂ gradient coating gradually increases from 13.75 to 27.75 nm, and the crystal structure is a face-centered cubic (FCC). The main crystal orientation faces are (111) and (200), while the (200) face exhibited a stronger preferred orientation. Compared with the Ni coating by scanning electron microscopy, the surface morphology of double pulse fabricated Ni/ZrO₂ gradient coating was revealed as being smoother, denser, and having fewer pores, and the crystal particle size distribution became narrow. X-ray photoelectron spectroscopy (XPS) shows that the chemical binding states of elements Ni and Zr have been altered. The binding energies of $2p_{3/2}$ and $2p_{1/2}$ for Ni have been increased, resulting in a higher electron donor state of Ni. The binding energy of $3d_{5/2}$ and $3d_{3/2}$ of $Zr^{4+}$ in ZrO₂ is decreased, thus becoming better electron acceptors. Chemical bonding has been formed at the Ni/ZrO₂ interface. This study demonstrated that double pulse electrodeposition is a promising fabrication method for functional gradient coatings for high temperature applications.

**Keywords:** gradient; double pulse; corrosion resistance; high temperature

---

## 1. Introduction

With the rapid development of the aviation, aerospace, power supply and sailing, crucial demands have emerged on functional materials under specific environments [1]. Metal matrix composite (MMC) coatings have been widely applied in many industries due to its high temperature resistance, oxidation resistance, high hardness and wear resistance as compared to pure metal coatings [2–8]. However, it has been shown that the ceramic particles incorporated in the composite coatings deteriorate the adhesion of the coatings to the substrate which, as a result, reduces their wear resistance significantly. To solve this problem, functional gradient materials (FGM) have been produced. The gradient and continuous evolution of the composition and structure of FGM accompany without the presence of phase interfaces, which lead to gradient variation of properties and functionalities along the direction of thickness. For implementing the excellent heat-resisting and antioxidative properties of FGM, many types of oxides are always employed, and ZrO₂ has been widely recognized as an appropriate inorganic component due to its high boiling point, high melting point, and low thermal expansion coefficient.

Many techniques for fabricating nanometric FGM, including physical and chemical vapor depositions, powder metallurgy, plasma spray, self-propagating [9,10] and in particular, electrodeposition, have been studied. Compared with the other approaches, electrodeposition can be readily operated under ambient conditions without sophisticated requirements on equipment and controlling [11]. To date, preparation of FGM via electrodeposition has been moderately researched in direct current deposition (DC) and pulse electrodeposition [12–14]. For example, Wang et al. [12] prepared the $Ni/ZrO_2$ FGM via DC electrodeposition; Yao et al. [13] reported the nano-Ni-W FGM by gradually increasing the bath temperature and current density with DC electrodeposition; Partho Sarker et al. [14] prepared the $ZrO_2(YSZ)-Al_2O_3$ FGM which adopted the method of DC electrodeposition. In contrast with the direct current deposition technique, double pulse electrodeposition could smooth the coating surface by dissolving the burrs during the reverse pulses. Both hydrogen brittleness and internal stress can be effectively reduced, resulting in better wear and corrosion resistance of the coating layers. Ge et al. [15] compared the properties of Ni coating layers via DC electrodeposition and double pulse electrodeposition and found that the double pulse electrodeposited coating layer is denser and more uniform, with better anticorrosion performance than that of DC electrodeposition. Similarly, Ma and coworkers [16] prepared the Ni/SiC composite coating via direct current deposition and double pulse electrodeposition. The results showed that double pulse electrodeposition offers considerably improved corrosion resistance of the coating layer than that prepared using double pulse plating. Consequently, we can rationally speculate that the double pulse electrodeposition might be an appropriate choice for preparing FGM.

In the present work, nano-grain $Ni/ZrO_2$ gradient coating and Ni coating was fabricated by double pulse electrodeposition. The surface morphology and microstructure of $Ni/ZrO_2$ gradient coating and Ni coating were compared at different high temperatures. The high temperature corrosion resistance of $Ni/ZrO_2$ gradient coating was characterized by oxidation weight increase experiments and electrochemistry. The purpose of this study was to investigate the effect of $ZrO_2$ nanoparticles on the high temperature corrosion resistance of $Ni/ZrO_2$ gradient coating, and finally provide a new process for preparing $Ni/ZrO_2$ gradient coating.

## 2. Materials and Methods

### 2.1. Fabrication of Ni/ZrO$_2$ Gradient Coatings

Nickel plate (160 × 80 × 8 mm, purity 99.99%) and stainless steel SUS201 (1Cr17Mn6Ni5N) (90 × 60 × 0.5 mm) were used as anode and cathode respectively. The cathode pretreatment process is depicted in Figure 1. The composition of the nickel plating solution and main plating parameters are listed in Table 1. The digital electroplating equipment of SMD-30 (Tai Shun Electroplating, Handan, China) was selected as the double pulse electroplating power supply. The experimental parameters are listed below: positive current duty cycle was 10%, positive cycle was 100 ms, positive current was 1.5 A/dm$^2$, reverse current duty cycle was 10%, reverse cycle was 20 ms, positive current was 0.2 A/dm$^2$. The waveform is given in Figure 2: TF is a set of forward pulse working times, TF = $n$T ($n \geq 1$); TR is a set of reverse pulse working times, TR = $-n$T ($n \geq 1$), TF + TR is a cycle of forward and reverse pulse commutation (generally TF > TR). The experimental operation parameters were: pH value is between 3.9–4.2, solution temperature was at 50 °C, stirring speed was 300 r/min, a certain amount of $ZrO_2$ powder was added into the plating solution in several times, and 16 orthogonal experiments were performed.

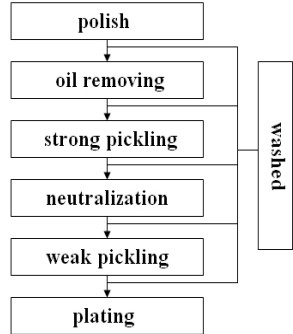

**Figure 1.** The flow chart of substrate pre-treatment.

**Table 1.** Formula of plating solution and operating conditions.

| Composition | Condition |
|---|---|
| $NiSO_4 \cdot 6H_2O$ | 250–300 g·L$^{-1}$ |
| $NiCl_2 \cdot 6H_2O$ | 40–50 g·L$^{-1}$ |
| $H_3BO_3$ | 30 g·L$^{-1}$ |
| $C_7H_5O_3NS$ (Saccharin) | 0.5 g·L$^{-1}$ |
| $C_{12}H_{25}SO_4Na$ (SDS) | 0.1 g·L$^{-1}$ |
| $ZrO_2$ (mean grain size:20 nm) | 20 g·L$^{-1}$ |
| positive current density | 1.0 A·dm$^{-2}$ |
| time | 120 min |

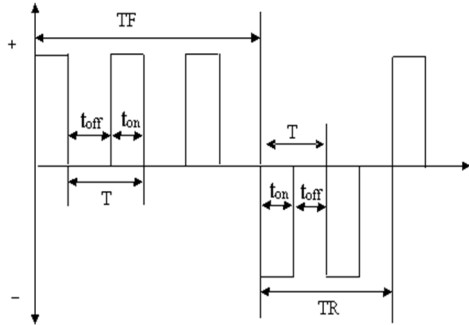

**Figure 2.** Waveform diagram of reverse pulse power.

## 2.2. Coatings Characterization

X-ray diffraction (XRD) (D8-Focus, Bruker, Billerica, MA, USA) analysis was carried out to determine the phase and crystal structure of the prepared coatings. The crystal size of the coatings was calculated according to the Scherrer formula. A scanning electron microscope (SEM) (FEI Quanta 450, Hillsboro, OR, USA) was used to characterize the surface morphology of the coatings in the corrosion test. The element composition of the coatings was analyzed by an energy dispersive spectrometer (EDS) (FEI, Hillsboro, OR, USA).

The linear polarization scans via conventional triple electrode system (CHI 660C, electrochemical workstation, Shanghai, China) were conducted at room temperature in 3.5 wt % NaCl aqueous solution to evaluate the corrosion behavior of the coatings. The 3 electrodes were an auxiliary electrode (Platinum), reference electrode (saturated calomel electrode), and a working electrode (coating with an area of 1 cm$^2$).

X-ray photoelectron spectroscopy (XPS) (VG Multilab 2000, Thermo, Waltham, MA, USA) was used to reveal the chemical state of each element in the coating, with Al Kα as the excitation source and the working pressure was $2 \times 10^{-6}$ Pa. Due to the inevitable C pollution as a result of the vacuum



pump oil evaporation in the vacuum chamber, the XPS peaks were calibrated with the binding energy of C 1*s* (284.6 eV).

## 3. Results

### 3.1. Cross-section Morphology and Element Distribution of Ni/ZrO₂ Gradient Coating

The cross-section morphology can be seen from Figure 3a: the color near the substrate is darker and the structure is denser; the lower side is closer to the coating surface, where more $ZrO_2$ nanoparticles have been added, and exhibits a brighter color and larger holes. The EDS mapping in Figure 3b shows the zirconium element distribution of the cross-sectional surface. It can be concluded from Figure 3 that the content of $ZrO_2$ particles gradually increases from top to bottom. The EDS quantitative results in Figure 4 clearly reveals this tendency:

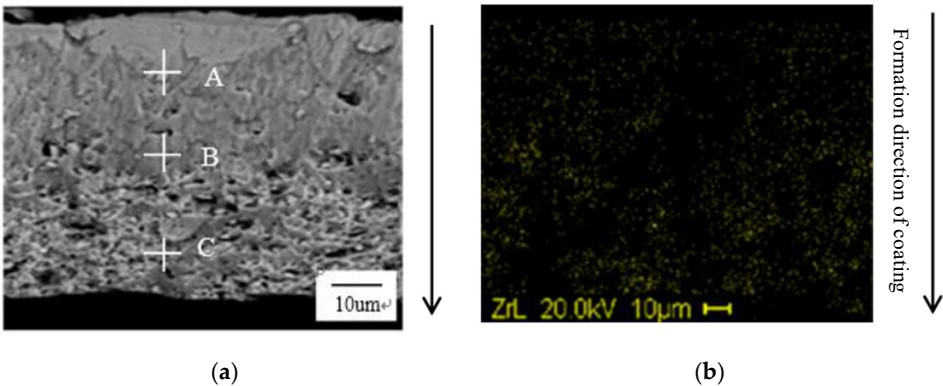

**Figure 3.** SEM image and element distribution mapping of cross-sectional Ni/ZrO₂ gradient coating. (**a**) Cross-section morphology; (**b**) Zr element distribution diagram.

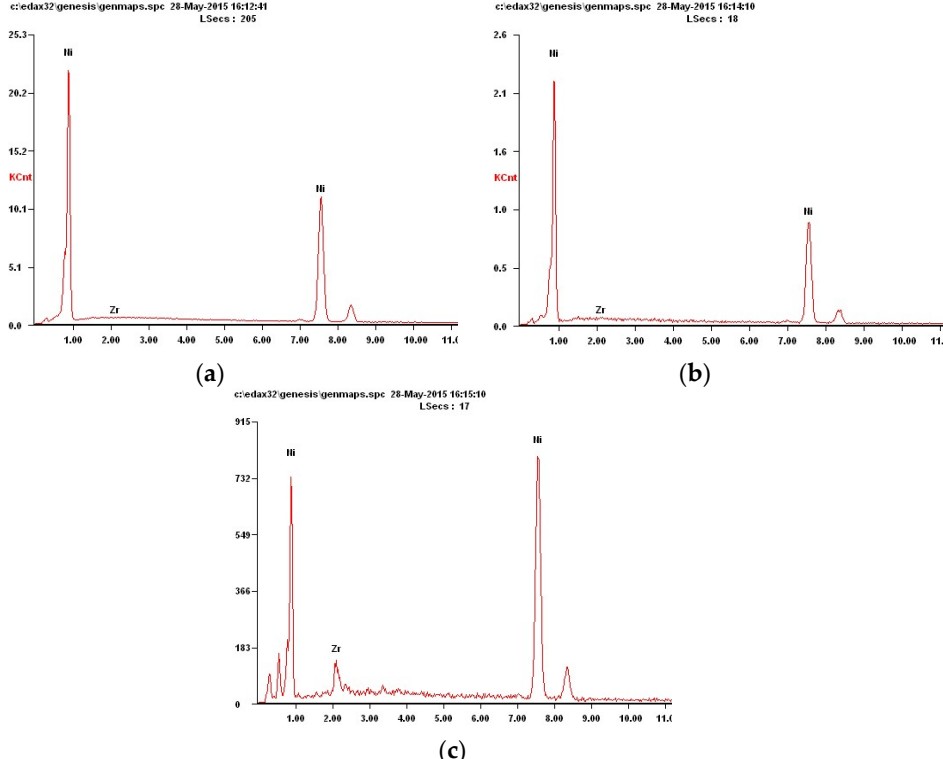

**Figure 4.** Point scan element curves of Ni/ZrO₂ gradient coating. (**a**) EDS plot at point A; (**b**) EDS plot at point B (**c**) EDS plot at point C.

Table 2 indicates that the content of $ZrO_2$ gradually increases from 12.39% to 34.99% from point A to point C. The combination of the above analysis proves the successful preparation of $Ni/ZrO_2$ gradient coating by double pulsed electrodeposition.

**Table 2.** The element content of points A, B and C.

| Element | A | | B | | C | |
|---|---|---|---|---|---|---|
| | **wt %** | **at %** | **wt %** | **at %** | **wt %** | **at %** |
| NiK | 87.67 | 91.65 | 72.31 | 80.22 | 65.01 | 74.27 |
| ZrK | 12.39 | 8.35 | 27.69 | 19.78 | 34.99 | 25.73 |

### 3.2. Analysis of Orthogonal Experimental Data

The XRD spectra of $Ni/ZrO_2$ gradient coatings prepared by orthogonal experiments have been carried out to identify the crystal structure, as shown in Figure 5, and the average grain size as calculated by Scherrer formula is shown in Table 3:

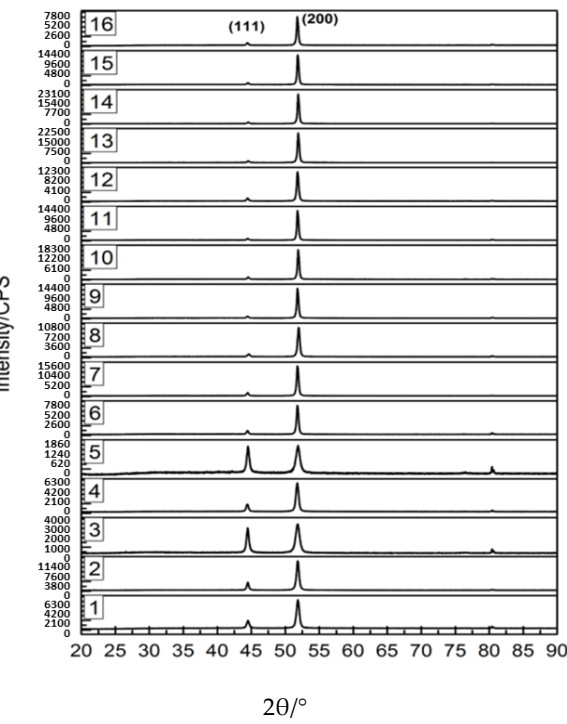

**Figure 5.** XRD pattern of $Ni/ZrO_2$ gradient coatings.

**Table 3.** Average grain size of $Ni/ZrO_2$ gradient coatings.

| Number | 1 | 2 | 3 | 4 | 5 | 6 | 7 | 8 |
|---|---|---|---|---|---|---|---|---|
| Average grain size (nm) | 17.2 | 20.6 | 14.6 | 18.35 | 13.75 | 23.5 | 23.65 | 26.4 |

| Number | 9 | 10 | 11 | 12 | 13 | 14 | 15 | 16 |
|---|---|---|---|---|---|---|---|---|
| Average grain size (nm) | 26 | 27.75 | 27.7 | 23.45 | 23.6 | 26.1 | 26.9 | 24.55 |

It can be seen from Figure 5 that the 16 groups of $Ni/ZrO_2$ gradient coatings prepared by orthogonal experiments have the main crystal face orientations of (111) and (200) planes, and the orientation advantages of (200) planes are more obvious. Table 3 indicates that the average size of the crystal is between 13.75 and 27.75 nm.

### 3.3. Surface Morphology of Ni/ZrO$_2$ Gradient Coating at Different Temperatures

The surface morphology of the pure Ni coating and the Ni/ZrO$_2$ gradient coating was tested at 25 °C. The results are shown in Figure 6.

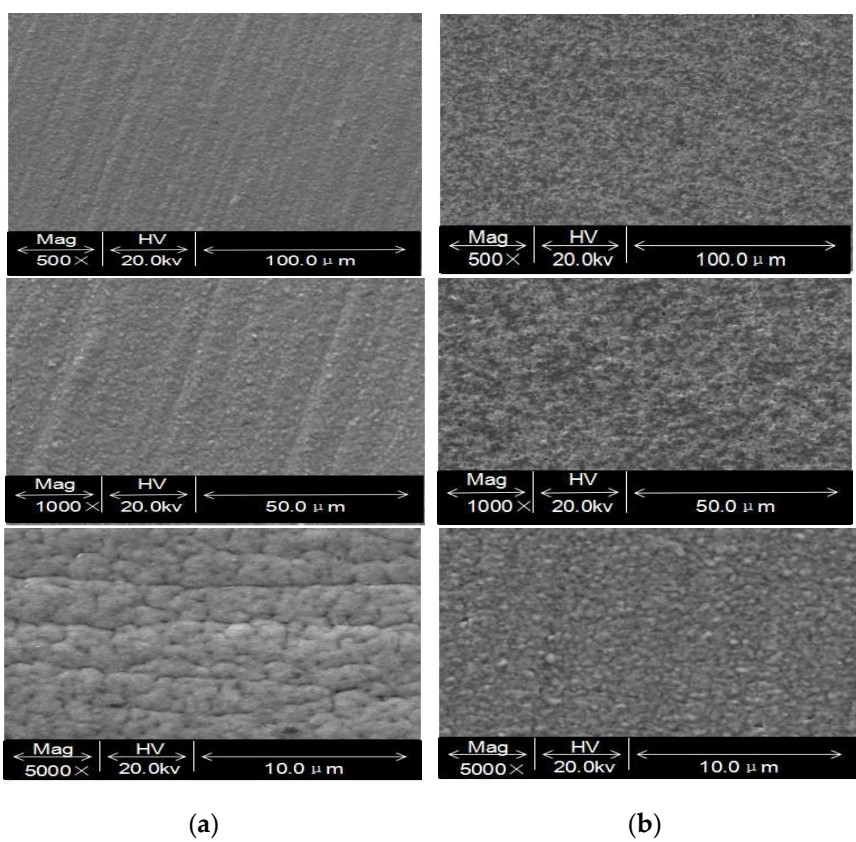

(**a**)                                        (**b**)

**Figure 6.** SEM image of pure nickel coating and Ni/ZrO$_2$ gradient coating under different magnification: (**a**) Ni coating, (**b**) Ni/ZrO$_2$ gradient coating.

Figure 6 represents the surface morphology of the Ni coating surface and Ni/ZrO$_2$ gradient coating, respectively. We found that the Ni/ZrO$_2$ gradient coating is smoother and more compact compared with the Ni coating. In the SEM image magnified 500 times, the surface of the Ni coating appeared slightly depressed, and revealed some fine textures, while the Ni/ZrO$_2$ gradient coating was smoother and denser, and almost with an absence of defects. At 1000 times, the texture of the Ni coating was deepened and the surface was uneven, and the Ni/ZrO$_2$ functional gradient coating was still very smooth and dense. At 5000 times, the surface of the Ni coating became coarse and over-thick, the particles were coarse, with obvious porosity, and the depression appeared, while the Ni/ZrO$_2$ gradient coating was still flat, dense, uniform in size without obvious defects.

Figure 7 shows the surface morphology of the Ni coating and Ni/ZrO$_2$ gradient coating after high temperature oxidation at 400–800 °C for 12 h. At 400 °C, few corrosion holes have appeared on the surface of Ni coating, while the surface of Ni/ZrO$_2$ gradient coating stays smooth without holes. After 500 °C, the corrosion holes of Ni coating increased, and only small corrosion holes were generated in Ni/ZrO$_2$ gradient coating. After 600 °C, the corrosion holes of Ni coating expanded and small corrosion holes appeared in Ni/ZrO$_2$ gradient coating. After 700 °C, there are a large number of corrosion holes on the Ni coating surface, and the surface structure is relatively loose; the corrosion holes of Ni/ZrO$_2$ gradient coatings only slightly increase. After 800 °C, the surface of Ni coatings is highly porous and the corrosion holes of Ni/ZrO$_2$ gradient coatings increase just a little bit. At high temperature, the nano-ZrO$_2$ particles in the Ni/ZrO$_2$ functional gradient coating provide a large number of preferential nucleation sites for oxide growth, promoting the nucleation and growth of the

protective oxide film, preventing oxygen diffusion, which significantly reduces the oxidation rate of $Ni/ZrO_2$ gradient coatings and further improves the high temperature corrosion resistance of $Ni/ZrO_2$ gradient coatings [17]. For Ni coatings, oxygen molecules diffuse into the coatings efficiently before the protective film has been formed. $Ni^{2+}$ moves along the Ni/NiO interface into the NiO layer by vacancy defect, increasing the diffusion channels for $Ni^{2+}$ which increased the oxidation rate, resulting in a large number of vacancy defects and created holes. The reason why $Ni/ZrO_2$ gradient coating has good high temperature corrosion resistance can be ascribed to the following reasons. Firstly, the uniform dispersion of nano-$ZrO_2$ particles in $Ni/ZrO_2$ functional gradient coating promotes the formation of a protective oxide film and hinders the diffusion of oxygen in the $Ni/ZrO_2$ functional gradient coating, so the oxidation rate of the $Ni/ZrO_2$ functional gradient coating has reduced. Secondly, the nano-$ZrO_2$ particles in the coating are uniformly dispersed both inside the grains and grain boundaries of the matrix metal, and the tight binding between $ZrO_2$ and the matrix metal reduces the effective area between the matrix metal Ni and the oxidizer. This further prevents the short-circuit diffusion of $Ni^{2+}$ on the surface, grain boundaries, and dislocations, reducing the diffusion channels of $Ni^{2+}$ and thus prevents the growth of grains during high temperature corrosion, leading to the improvement of the high temperature corrosion resistance of $Ni/ZrO_2$ gradient coating.

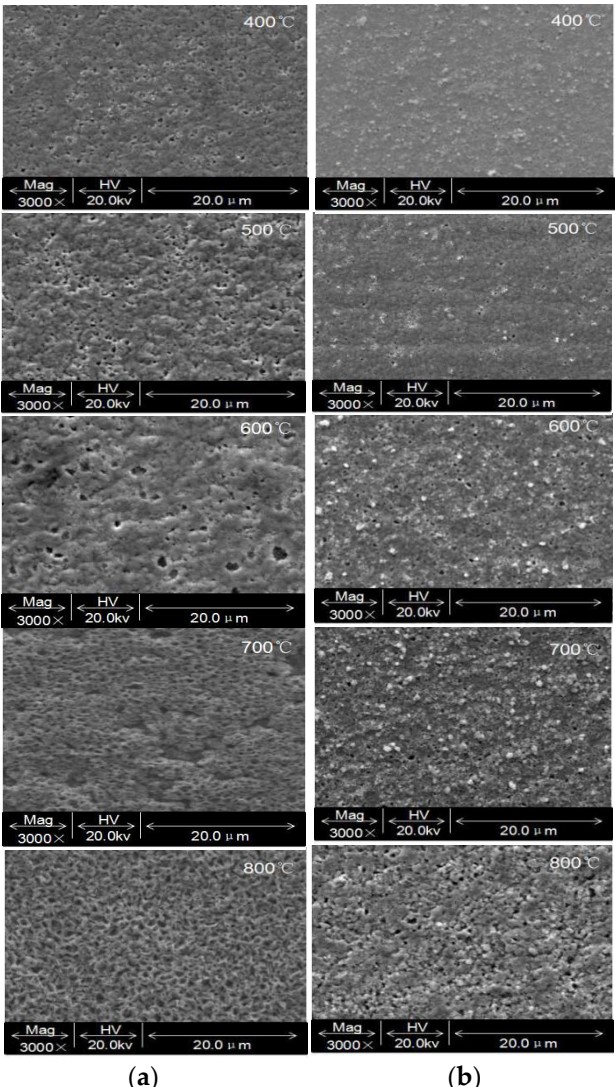

**Figure 7.** SEM image of Ni coating and $Ni/ZrO_2$ gradient coating after 12 h high temperature treatment: (**a**) Ni coating, (**b**) $Ni/ZrO_2$ gradient coating.

### 3.4. Structure of Ni/ZrO₂ Gradient Coating at Different Temperatures

The structure of Ni/ZrO$_2$ gradient coating was characterized after 12 h of continuous high temperature at 400, 500, 600, 700 and 800 °C respectively, and the structural changes were analyzed and compared with the spectra at 25 °C, as shown in Figure 8:

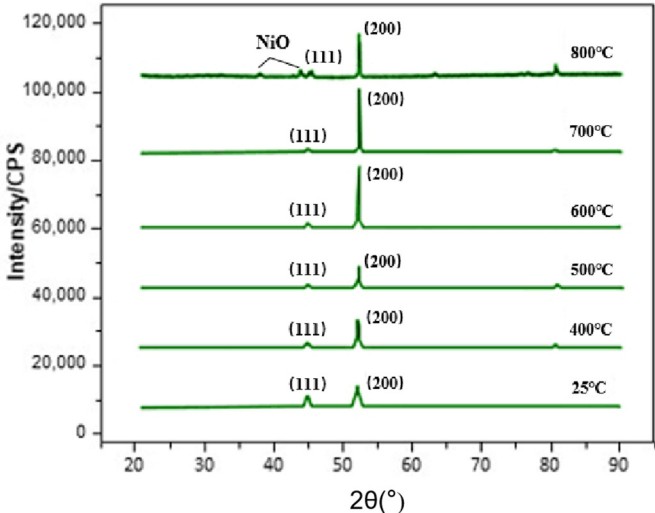

**Figure 8.** XRD pattern of Ni/ZrO$_2$ gradient coating after high temperature treatment.

It can be seen from Figure 8 that the preferred orientation planes of Ni/ZrO$_2$ gradient coatings at different high temperatures are (111) and (200) planes, and the orientation advantage of (200) planes is more preferable. At 800 °C, NiO is partially oxidized as characteristic NiO peaks appear. Comparison of the spectra at 25 (normal temperature), 400, 500, 600, 700, and 800 °C shows that there is no significant change in the structure of the coating structure, and the preferred orientations of the crystal planes are (111) and (200).

### 3.5. Electrochemical Corrosion Performance of Ni/ZrO₂ Gradient Coatings after High Temperatures Treatment

It can be seen from Figures 9 and 10 and Tables 4 and 5 that as the temperature rises from 400 to 800 °C, the self-corrosion potential ($E_{corr}$) and polarization resistance ($R_p$) of the Ni coating and Ni/ZrO$_2$ gradient coating tend to decrease, and the corrosion resistance of Ni/ZrO$_2$ gradient coating is better than that of Ni coating. After the Ni plating layer is oxidized, nickel oxide is the main product. This oxide is a P-type semiconductor oxide. Vapor defects are the main routes of diffusion. During the oxidation process, a large number of vacancy defects and some holes are generated, leading to of increase in the grain size of the coating, which facilitates the corrosion. The Ni/ZrO$_2$ gradient coating can protect the substrate due to its fine structure, which inhibits the diffusion of external atoms into the coating and slows down the corrosion process. Because the nano-ZrO$_2$ particles are uniformly distributed in the Ni/ZrO$_2$ gradient coating, the corrosion pits on the surface of the Ni/ZrO$_2$ gradient coating are uniformly dispersed, which effectively prevents the expansion of the corrosion pits.

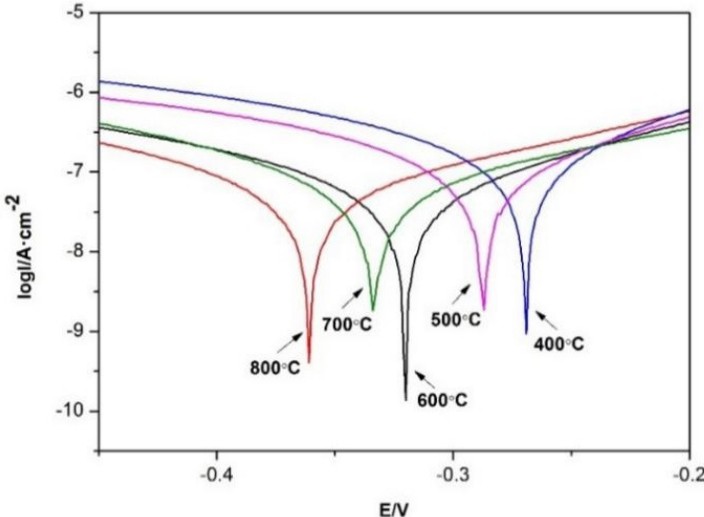

**Figure 9.** Tafel polarization curves of Ni/ZrO$_2$ gradient coatings after treatments at high temperatures.

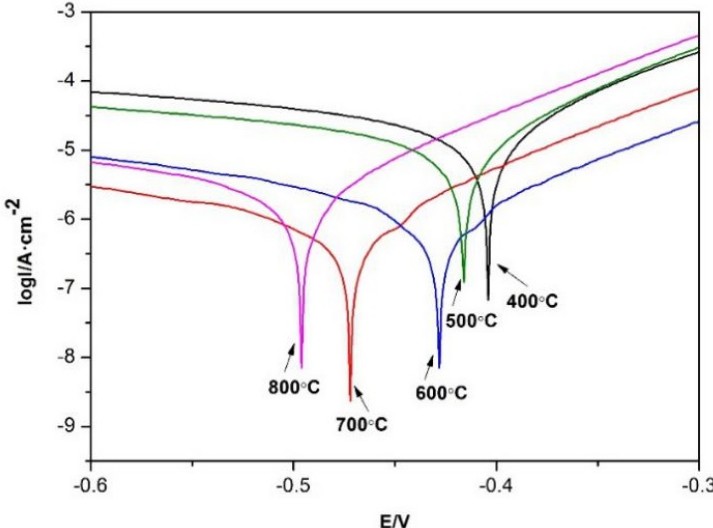

**Figure 10.** Tafel polarization curves of Ni coatings after treatments at high temperatures.

**Table 4.** $E_{corr}$, $I_{corr}$ and $R_p$ values of Ni/ZrO$_2$ gradient coatings after treatments at high temperatures.

| Temperature/°C | $E_{corr}$/V | $I_{corr}$/A·cm$^{-2}$ | $R_p$/Ω·cm$^2$ |
|---|---|---|---|
| 400 | −0.269 | $5.92 \times 10^{-7}$ | 62,659 |
| 500 | −0.287 | $7.24 \times 10^{-7}$ | 46,568 |
| 600 | −0.312 | $1.37 \times 10^{-6}$ | 28,780 |
| 700 | −0.3336 | $1.78 \times 10^{-6}$ | 18,427 |
| 800 | −0.362 | $2.14 \times 10^{-6}$ | 16,970 |

**Table 5.** $E_{corr}$, $I_{corr}$, and $R_p$ values of Ni coatings after treatments at high temperatures.

| Temperature/°C | $E_{corr}$/V | $I_{corr}$/A·cm$^{-2}$ | $R_p$/Ω·cm$^2$ |
|---|---|---|---|
| 400 | −0.401 | $8.91 \times 10^{-8}$ | 6282 |
| 500 | −0.418 | $9.85 \times 10^{-8}$ | 5793 |
| 600 | −0.425 | $1.09 \times 10^{-9}$ | 5425 |
| 700 | −0.468 | $2.75 \times 10^{-9}$ | 4468 |
| 800 | −0.497 | $7.02 \times 10^{-9}$ | 3676 |

### 3.6. Oxidation Weight Gain

Ni coating and Ni/ZrO$_2$ gradient coating were subjected to a high temperature oxidation weight gain test at 600 °C for 8–24 h. The change of oxidation weight gain is shown in Table 6. The oxidation weight gain curves of the two coatings were fitted and the results were obtained and are shown in Figures 11 and 12.

**Table 6.** Weight gain of Ni coating and Ni/ZrO$_2$ gradient coating at 600 °C for 8–24 h.

| Time/h | Ni/g·cm$^{-2}$ | Ni/ZrO$_2$ Gradient Plating/g·cm$^{-2}$ |
|--------|--------|--------|
| 8 | 5.6 | 0.6 |
| 12 | 5.9 | 0.9 |
| 16 | 12.6 | 1.5 |
| 20 | 26.1 | 3.6 |
| 24 | 28.9 | 4.8 |

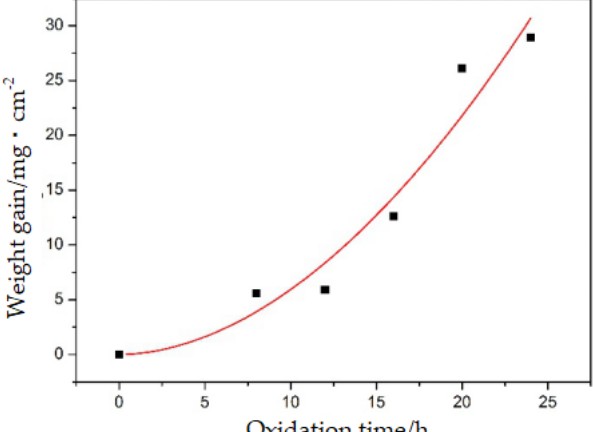

**Figure 11.** Weight gain curves of Ni coating.

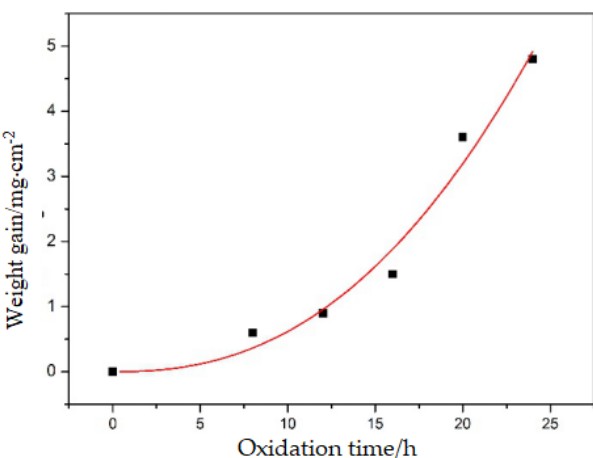

**Figure 12.** Weight gain curves of Ni/ZrO$_2$ coating.

It can be seen from Figures 11 and 12 that the oxidation weight increase tendency of Ni plating and Ni/ZrO$_2$ gradient plating both increases slowly at the initial stage and increases rapidly later as oxidation time increases. However, the weight gain of Ni/ZrO$_2$ gradient coatings is significantly lower than that of Ni coatings. Ni/ZrO$_2$ gradient coatings exhibit an improved high temperature oxidation resistance. The fitting of the curves shows that the oxidation law of the surface of the two coatings obeys the power function relationship, and the relationship between the Ni coatings is: $W = 0.07893t^{1.876}$, and the correlation coefficient is 0.93131; the relation of Ni/ZrO$_2$ coating is: $W = 0.00266t^{2.368}$, and the

correlation coefficient is 0.97143, respectively. Comparing the correlation coefficient between the two further validates the enhanced high temperature oxidation resistance of Ni/ZrO$_2$ gradient coatings.

### 3.7. XPS Analysis

XPS analysis was conducted on the prepared Ni coating, Ni/ZrO$_2$ gradient coating and pure ZrO$_2$; the results are shown in Figure 13. In order to determine the chemical environment of the Ni element in each coating, high-resolution XPS scanning was performed. For comparison, the Ni spectra of the Ni coating and the Ni/ZrO$_2$ gradient coating were combined in a single figure.

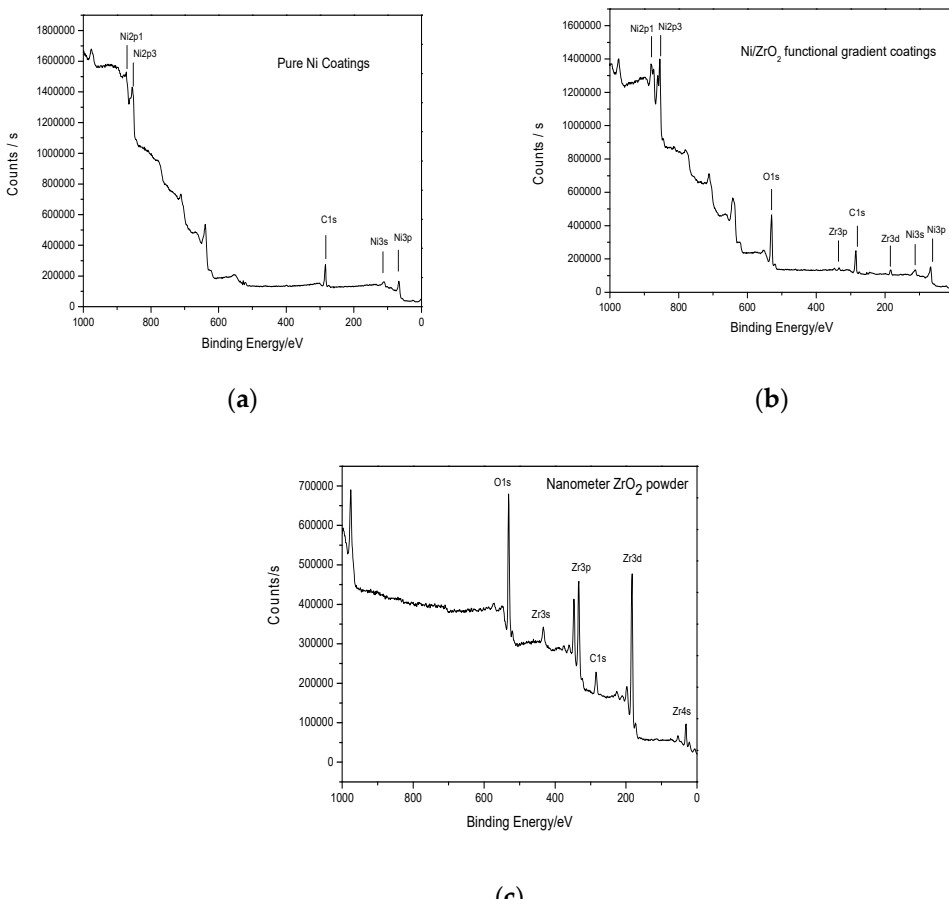

**Figure 13.** XPS spectrum of ZrO$_2$. (**a**) Pure Ni coatings; (**b**) Ni/ZrO$_2$ functional gradient coatings; (**c**) Nanometer ZrO$_2$ powder.

According to the standard spectrum manual, it can be seen that the XPS peaks in Figure 14 are from the $2p_{3/2}$ and $2p_{1/2}$ peaks of Ni. It was given in Table 7 that the binding energies of the $2p_{3/2}$ and $2p_{1/2}$ peaks of the pure nickel plating Ni are 852.35 and 870.27 eV, which are very close to the standard values of 852.3 and 869.7 eV. Based on the binding energy of pure nickel, the $2p_{3/2}$ and $2p_{1/2}$ binding energies of Ni in the Ni/ZrO$_2$ functional gradient coating were calculated to be shifted positively by 0.79 and 1.37 eV, compared to those of the Ni coating. This shift indicates that the introduction of ZrO$_2$ modifies the chemical bonding state of Ni. The binding energy of $2p_{3/2}$ and $2p_{1/2}$ of Ni increases, so Ni is in a higher electron donor state. In order to determine the chemical environment of ZrO$_2$ in each coating, the ZrO$_2$ spectra of powdered ZrO$_2$ and Ni/ZrO$_2$ gradient coatings are combined in a single figure. The results are shown in Figure 15.

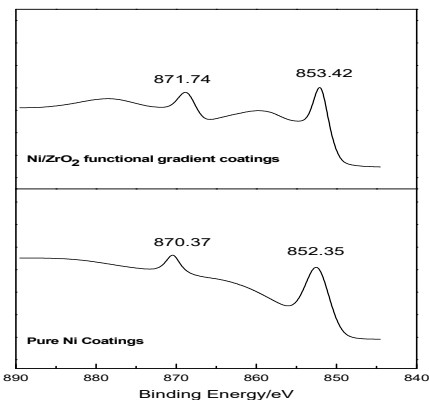

**Figure 14.** XPS spectrum of Ni element.

**Table 7.** *2p* electron bonding energies of each Ni coating.

| Sample | $2p_{3/2}$ | | $2p_{1/2}$ | |
|---|---|---|---|---|
| | Eb (eV) | △Eb (eV) | Eb (eV) | △Eb (eV) |
| Standard Ni | 852.3 | – | 869.7 | – |
| Pure Ni Coating | 852.35 | 0 | 870.37 | 0 |
| Ni/ZrO$_2$ functional gradient coatings | 853.14 | 0.84 | 871.74 | 1.37 |

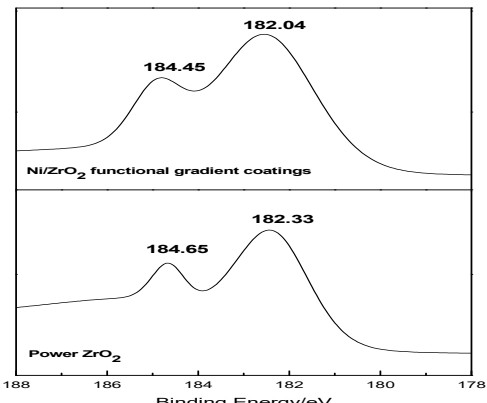

**Figure 15.** XPS spectrum of Zr element.

According to the standard spectrum manual, the XPS peaks in Figure 15 are derived from the $3d_{5/2}$ and $3d_{3/2}$ peaks of Zr$^{4+}$ in ZrO$_2$, respectively. Table 8 shows that the binding energies of $3d_{5/2}$ and $3d_{3/2}$ of Zr$^{4+}$ in ZrO$_2$ are 182.33 and 184.65 eV, which just show the minor shifts from the standard values of 182.2 and 184.5 eV. Based on the binding energy of ZrO$_2$ powder, it can be calculated that the binding energies of $3d_{5/2}$ and $3d_{3/2}$ of Zr$^{4+}$ in ZrO$_2$ in Ni/ZrO$_2$ gradient coating are negatively shifted by 0.29 and 0.2 eV, compared to ZrO$_2$ powder. The chemical binding state of Zr has been changed, with the binding energy of $3d_{5/2}$ and $3d_{3/2}$ of Zr$^{4+}$ in ZrO$_2$ were reduced, which tends to get electrons.

**Table 8.** *3d* electron binding energy of Zr in each coating.

| Sample | $3d_{5/2}$ | | $3d_{3/2}$ | |
|---|---|---|---|---|
| | Eb (eV) | △Eb (eV) | Eb (eV) | △Eb (eV) |
| Standard ZrO$_2$ | 182.2 | – | 184.5 | – |
| ZrO$_2$ powders | 182.33 | 0 | 184.65 | 0 |
| Ni/ZrO$_2$ functional gradient coating | 182.04 | −0.29 | 184.45 | −0.2 |

From the results of XPS analysis, it is known that in the gradient coating, chemical bonding has been formed between the particles and the matrix metal. Ni increases the binding energy and tends to donate electrons; Zr decreases the binding energy and tends to get electrons. The electronegativity data of the three elements Ni, Zr and O are: Ni: 1.91; Zr: 1.33; and O: 3.44. It is known from the electronegativity of the two that the possibility of direct bonding between Ni and Zr is low. $ZrO_2$ is a compound mainly composed of ionic bonds [18]. The bonding energy of Zr and O is not particularly large, and for nano-$ZrO_2$, the unsaturated chemical bonds of O are rich on the surface, so it should be the 3$d$ orbital of Ni atom, which interacts with the coordination electrons of highly electronegative O in $ZrO_2$. This bonding weakens the shielding effect of the outer electrons on the inner electrons of Ni, and increases the binding energy of Ni 2$p$. For $ZrO_2$, because some of the electrons of O participate in the bonding with Ni, the bonding ability of O to Zr is lower, the outer electrons of the original Zr bonded to O are released, and the outer electron density increases. The shielding effect of Zr outer layer electrons to inner layer electrons is enhanced, resulting in the reduction of the binding energy of Zr inner layer electrons.

## 4. Conclusions

In this work, we successfully prepared nano-Ni/$ZrO_2$ gradient functional coatings on the surface of stainless steel by double pulse electrodeposition at normal temperature and pressure. Compared with other methods, this method has the characteristics of simplicity, safety, and easy operation. Especially on the basis of applying reverse pulse current and without changing the composition of the solution, a bright, flat, uniform and dense nano-gradient coating is obtained by this method. At the same time, research shows that, compared with ordinary Ni coatings, the composition and structure of Ni/$ZrO_2$ gradient coatings have not changed at 400–800 °C, and the high temperature corrosion performance is significantly better than ordinary Ni coatings, which has changed the original method relying on the addition of additives to improve the performance of the coating. In general, this research is helpful in providing new ideas and approaches for solving the environmental protection problems of additives with strong toxicity and difficult treatment of waste liquid in actual production. The process method has a low cost and has good environmental protection application prospects.

**Author Contributions:** Methodology, W.G.; validation, T.H., M.W. and J.L.; writing-review and editing, W.G. and T.S.; writing-original draft preparation, W.G. and T.S.; supervision, W.G.; data curation, T.H, M.W. and J.L. All authors have read and agreed to the published version of the manuscript.

**Funding:** This work was supported by National Nature Science Foundation of China (50801057).

**Conflicts of Interest:** The authors declare no conflict of interest.

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
