# Peer review of "Nano-Grain Ni/ZrO2 Functional Gradient Coating Fabricated by Double Pulses Electrodeposition with Enhanced High Temperature Corrosion Performance"

_coatings, doi:10.3390/coatings10040332_

Round 1

Reviewer 1 Report

The paper presents useful information. However, some aspects need improvement. It is difficult for a native English reader to understand the paper unless the quality of English is improved. I also suggest that the corrosion data for the substrate is also presented for comparison.

Author Response

Comments and Suggestions for Authors:

The paper presents useful information. However, some aspects need improvement. It is difficult for a native English reader to understand the paper unless the quality of English is improved. I also suggest that the corrosion data for the substrate is also presented for comparison.

Dear reviewers 1:

For your comments and suggestions.I have improved the quality of English and compared the corrosion data for Ni coating and Ni / ZrO2 gradient coating.

Reviewer 2 Report

In the manuscript, the authors describe preparation of a Ni/ZrO2 FGM.

The introduction is sufficient. However, in the second paragraph, the authors can also mention some other methods FGM by which can be fabricated, such as certain methods of intensive plastic deformation which impart gradient structure formation (e.g. Lenka Kunčická, Radim Kocich, Karel Dvořák, Adéla Macháčková, Rotary swaged laminated Cu-Al composites: Effect of structure on residual stress and mechanical and electric properties, Materials Science and Engineering: A, 742, 2019, 743-750; and Radim Kocich, Lenka Kunčická, Petr Král, Pavel Strunz, Characterization of innovative rotary swaged Cu-Al clad composite wire conductors, Materials & Design, 160, 2018, 828-835).

Also, the authors could provide more information about prospective applications of the proposed material.

In Fig. 1, the authors should denote more clearly what is the substrate and the growth direction of the coating.

Author Response

Comments and Suggestions for Authors:

The introduction is sufficient. However, in the second paragraph, the authors can also mention some other methods FGM by which can be fabricated, such as certain methods of intensive plastic deformation which impart gradient structure formation.

Also, the authors could provide more information about prospective applications of the proposed material.

In Fig. 1, the authors should denote more clearly what is the substrate and the growth direction of the coating.

Dear reviewers:

For the suggestions that “authors can also mention some other methods FGM by which can be fabricated” and “the authors could provide more information about prospective applications of the proposed material”. I have introduced in the article.

For the suggestions that“In Fig. 1, the authors should denote more clearly what is the substrate and the growth direction of the coating”.l have indicated the growth direction of the coating.

Round 2

Reviewer 1 Report

Thank you submitting the paper to 'Coatings'. The general content is of interest to the reader. However, some work is required before it could be accepted for publication. Please find some suggesting to improve the paper:

  1. More references are required in the introduction.  There are several papers on Co in Ni coatings. The area of composite coatings is growing and this needs to be reflected in the introduction.
  2. Grain size distribution of ZrO2 particles uses would be useful.
  3. Better micrographs in Figure 1 would be useful.
  4. Details of Tafel experiments are missing
  5. The assumption that similar Ecorr means similar corrosion behaviour is flawed. One needs to take the polarisability of the material into account. A difference of 50mV is not insignificant (please consider this in the paper).
  6. As the coatings are on a metallic substrate, please carry out control experiments on the substrate to get the baseline value
  7. Stainless steel should be defined- UNS number or type
  8. The discussion on corrosion and electrochemical behaviour should be improved.
  9. The mechanism proposed is plausible. However, charge density of Ni2+ is 134 C mm-3. This needs to be taken into account. The Ni2+ is hydrated by six water molecules. How many ZrO2 molecules are carried by a single Ni2+ ion?
  10. Conclusions must be improved to bring out the new findings.

Author Response

Dear reviewer, The manuscript you reviewed is the one I submitted earlier. You have not reviewed the revised manuscript. I have uploaded the revised manuscript. Please review it.

Reviewer 2 Report

All the previous comments have been sufficiently addressed.

Author Response

Dear reviewer,

     Thank you for your comments.

Round 3

Reviewer 1 Report

Thanks for submitting the revised version. The quality of work is excellent. Some English corrections are required. Please see the attached file.

Author Response

Point 1:The Fabrication and enhanced high temperature corrosion performance of nano-grain Ni/ZrO2 functionally gradient coating by bidirectional pulse electrodeposition

Response 1:Nano-grainNi/ZrO2 functional gradient coating fabricated by double pulses electrodeposition with enhanced high temperature corrosion performance

Point 2: The Ni/ZrO2 nanocomposite coating has the good application prospects on the surface of the gas turbine jet engines due to its high temperature resistance, oxidation resistance, high hardness and wear resistance [2 -8].

Respone 2:Metal matrix composite (MMC) coatings have been widely applied in many industries due to its high temperature resistance, oxidation resistance, high hardness and wear resistance as compared to the pure metal coatings[2-8]

Point 3: nano-grain Ni/ZrO2 gradient coating and Ni coating will be fabricated by bidirectional pulse electrodeposition. The surface morphology and microstructure of Ni/ZrO2 gradient coating and Ni coating were compared at different high temperatures. The high temperature corrosion resistance of Ni/ZrO2 gradient coating was characterized by oxidation weight increase experiment and electrochemistry.

Respone 3: nano-grain Ni/ZrO2 gradient coating and Ni coating will be fabricated by double pulse electrodeposition. The surface morphology and microstructure of Ni/ZrO2 gradient coating and Ni coating were compared at different high temperatures. The high temperature corrosion resistance of Ni/ZrO2 gradient coating was characterized by oxidation weight increase experiment and electrochemistry. The purpose of this study was to investigate the effect mechanism of ZrO2 nanoparticles on the high temperature corrosion resistance of Ni / ZrO2 gradient coating, and finally provide a new process for fabricating Ni / ZrO2 gradient coating

Point 4:X-ray diffraction (XRD) analysis was carried out to determine phase and crystal structure of the prepared coatings. The crystal size of the coatings was calculated according to Scherrer formula. (D8-Focus, Bruker, Germany).

Scanning electron microscope (SEM) was used to characterize the surface morphology of the coatings in corrosion test. The element composition of the coatings were analyzed by energy dispersive spectrometer (EDS) (FEI Quanta 450 , America).

Respone 4:X-ray diffraction (XRD) (D8-Focus, Bruker, Germany) analysis was carried out to determine phase and crystal structure of the prepared coatings. The crystal size of the coatings was calculated according to Scherrer formula. Scanning electron microscope (SEM) (FEI Quanta 450 , US) was used to characterize the surface morphology of the coatings in corrosion test. The element composition of the coatings were analyzed by energy dispersive spectrometer (EDS).

Point 5:The Ni / ZrO2 gradient coating was acted as a protective film due to its fine structure, which inhibits the diffusion of external atoms into the coating, slow down the corrosion process. Thus the size of the etch pits has been reduced. and the nano-ZrO2 particles are uniformly dispersed in the Ni / ZrO2 gradient coating. The ground distribution leads to the uniform dispersion of the corrosion pits on the surface of the Ni / ZrO2 gradient coating , which effectively prevents the growth of the corrosion pits.

Respone 5:The Ni / ZrO2 gradient coating can protect the substrate due to its fine structure, which inhibits the diffusion of external atoms into the coating and slows down the corrosion process.because the nano-ZrO2 particles are uniformly distributed in the Ni / ZrO2 gradient coating, the corrosion pits on the surface of the Ni / ZrO2 gradient coating are uniformly dispersed, which effectively prevents the expansion of the corrosion pits.
